# Cross-cultural evaluation of the French version of the Delusion Assessment Scale (DAS) and Psychotic Depression Assessment Scale (PDAS)

Isabelle Jalenques[1]*, Chloé Rachez[2], Urbain Tauveron Jalenques[3], Silvia Alina Nechifor[4], Lucile Morel[5], Florent Blanchard[5], Bruno Pereira[6], Sophie Lauron[7], Fabien Rondepierre[2], for the French DAS/PDAS group[¶]

1 Service de Psychiatrie de l'Adulte et Psychologie Médicale, Centre Mémoire de Ressources et de Recherche, Centre Hospitalier Universitaire Clermont-Ferrand, Université Clermont Auvergne, Clermont-Ferrand, France, 2 Service de Psychiatrie de l'Adulte et Psychologie Médicale, Centre Mémoire de Ressources et de Recherche, Centre Hospitalier Universitaire Clermont-Ferrand, Clermont-Ferrand, France, 3 Université Clermont Auvergne, Clermont-Ferrand, France, 4 Centre Hospitalier Spécialisé Sainte-Marie, Clermont-Ferrand, France, 5 Centre Hospitalier Spécialisé Sainte-Marie, Le Puy-en-Velay, France, 6 Direction de la Recherche Clinique et de l'Innovation, Centre Hospitalier Universitaire Clermont-Ferrand, Clermont-Ferrand, France, 7 Service de Psychiatrie de l'Adulte et Psychologie Médicale, Centre Hospitalier Universitaire Clermont-Ferrand, Clermont-Ferrand, France

¶ Membership of the French DAS/PDAS Group is provided in the Acknowledgments.
* ijalenques@chu-clermontferrand.fr

**Data Availability Statement:** All data are available at Mendeley: Jalenques, Isabelle; Tauveron-Jalenques, Urbain; Rachez, Chloé; Nechifor, Silvia

## Abstract

### Background

Major depressive disorder with psychotic features (MDDPsy), compared to nonpsychotic MDD, involves an increased risk of suicide and failure to achieve treatment response. Symptom scales can be useful to assess patients with MDDPsy. The aim of the present study was to validate French versions of the Delusion Assessment Scale (DAS) and Psychotic Depression Assessment Scale (PDAS).

### Methods

One hundred patients were included. The scales were filled out by psychiatrists. Data from participants who accepted a second interview were used for inter-judge reliability. The scalability and psychometric properties of both scales were assessed.

### Results

Data from 94 patients were used. Owing to low score variability between patients, the predefined threshold for scalability (≥0.40) was not reached for both scales. Factorial analysis of the DAS identified five factors, different from those of the original version. Five factors were also identified in the PDAS, of which two comprised items from the HDRS and the other three items from the BPRS. Floor and ceiling effects were observed in both scales, due in part to the construction of certain subscales. Unlike the PDAS, the DAS had good internal

Alina; Morel, Lucile; Blanchard, Florent; Pereira, Bruno; Lauron, Sophie; Rondepierre, Fabien (2020), "Cross-cultural evaluation of the French version of the Delusion Assessment Scale (DAS) and Psychotic Depression Assessment Scale (PDAS)", Mendeley Data, V1, doi: 10.17632/kw7m5f4sjv.1.

**Funding:** The authors received no specific funding for this work.

**Competing interests:** The authors have declared that no competing interests exist.

consistency. Multiple correlations were observed between the DAS dimensions but none between those of the PDAS. Both scales showed good inter-judge reliability. Convergent validity analyses showed correlations with HDRS, BPRS and CGI.

## Limitations

Inter-judge reliability was calculated from a relatively small number of volunteers.

## Conclusions

The good psychometric properties of the French versions of the DAS and PDAS could help in assessing MDDPsy, in particular its psychotic features, and hence improve response to treatment and prognosis.

## Introduction

Major depressive disorder with psychotic features (MDDPsy) is defined by the presence of delusions and/or hallucinations during the episode of major depression [1]. Studies have reported a prevalence rate of psychotic features in MDD ranging from 11 to 18.5% [2–4]. MDDPsy is characterized by a greater depressive symptom severity and a higher probability of melancholic features. In addition, patients with MDDPsy have a greater risk of suicide than nonpsychotic MDD patients [5, 6]. They also have greater probability of inpatient treatment. Most international guidelines recommend the combination of an antidepressant and an antipsychotic or electroconvulsive therapy for the treatment of MDDPsy [7–11]. However, MDDPsy patients are more likely not to achieve treatment response than nonpsychotic MDD patients [4]. MDDPsy is associated with greater psychosocial impairment [12] and decreased quality of life [13]. Combined assessment of psychotic characteristics and overall severity is crucial in MDDPsy [14].

Symptom scales can be useful tools for assessment of measurement-based care for the individual patient and for evaluating and monitoring outcomes in research studies. There exist validated scales that can help to assess patients suffering from MDDPsy [14] which could be more widely used if they were made available in the mother tongue of clinician-researchers. We decided therefore to select two scales among validated rating scales measuring psychotic depression, the Delusion Assessment Scale (DAS), designed specifically for the assessment of delusions in MDDPsy [15], and the Psychotic Depression Assessment Scale (PDAS), which assesses the overall severity of depressive and psychotic symptoms in MDDPsy [16] and has also shown promising results in detecting the occurrence of psychotic depression among patients with depressive disorders in general [17, 18]. The present study aimed to validate French versions of the DAS and PDAS after evaluation of their psychometric properties for use in measuring the severity of psychotic depression in MDDPsy patients.

## Methods

### Study design

The project was approved by the local Ethical Review Board (Comité de Protection des Personnes Sud-Est VI Clermont-Ferrand, IRB 00008526) and conducted according to Good Clinical Practice guidelines and the Declaration of Helsinki. A psychiatrist explained the aims and procedures to the patients and collected their consent to participate in the study. All patients received written information about the study and were entitled to refuse use of the data collected at any time.

The psychiatrist completed the DAS and PDAS scales after a routine interview with patients shortly after hospital admission. Data from patients who accepted an interview with a second psychiatrist were used for the inter-judge reliability assessment.

## Participants

Inpatients aged 18 or older with fluent command of French were recruited from the psychiatric units of the university hospital and the psychiatric hospital centres of Clermont-Ferrand and Le Puy en Velay. Inclusion criteria were diagnosis of major depressive disorder with psychotic features on the basis of DSM 5 criteria [1], a delusional idea as defined in DSM-5 with a rating ≥3 (delusion definitely present) on the delusional rating item of the Schedule for Affective Disorders and Schizophrenia [19] and a score of ≥2 on one of the DAS conviction items, and a score of ≥ 21 on the 17-item version of the Hamilton Depression Scale [20].

Exclusion criteria were a history of schizophrenia, schizoaffective disorder, delusional disorder, shared psychotic disorder, mania, dementia, history of drug or alcohol abuse in the past three months, taking medication likely to induce psychiatric symptoms or disorders or having an unstable or life-threatening medical illness, and meeting criteria for obsessive-compulsive disorder and body dysmorphic disorder.

## Instruments

**The DAS instrument.** The DAS assesses domains of delusional ideation in patients with MDDPsy. It expands the five single-item domains of the DDERS [21]—conviction, pressure, extension, bizarreness and disorganization—into a 14-item scale with each item rated on 1–3 anchor points. The highest score for each item corresponds to greatest disease severity. A fifteenth item was added to assess mood congruence, which corresponds to a single separate dimension. Rating was performed by an assessor after a standard clinical interview, observation of the patient and reading of comments made by the care team or close friends or family. An instruction manual is available to guide the interview and ratings. It is recommended to assess the severity of symptoms over the past week [15].

**Instrument translation.** The DAS was cross-culturally adapted from English into French following established guidelines [22], with the kind permission of the author of the original instrument (Barnett S. Meyers, M.D.). Forward translations were independently made by three psychiatrists fluent in English (CR, SN, IJ), and one bilingual translator, naive to the outcome measure (UTJ), all of whom had French as their mother tongue. A native English translator fluent in French (JW) with a medical background blinded to the original English version then made a backward translation. A multidisciplinary expert committee reviewed the process, compared source and target versions, and resolved discrepancies. Item-translation, semantic, idiomatic, experiential, and conceptual equivalents were discussed. Finally, the consensus target version was adopted by the committee as the pre-final cross-cultural adaptation (S1 File).

We then made an evaluation of the psychometric properties of DAS, given in detail below.

**The PDAS instrument.** The PDAS was constructed specifically to assess the severity of psychotic depression in patients with MDDPsy [16]. It is composed of six items from the Hamilton Depression Scale (HAM-D6) [20] and five items from the Brief Psychiatric Rating Scale (BPRS5) [23], both of which are publicly available. In addition, there exist validated French versions of these two scales [24, 25]. The total PDAS score is obtained by adding up the sum of the individual item scores after converting the BPRS item scores to a score between 0 and 4 with the formula (BPRS score– 1) x 2/3, and multiplying the score for item 13 on the Hamilton scale (general somatic) by 2 [16]. Rating is performed by the assessor on the basis of a clinical

interview and observation of the patient during the conversation. Guidelines recommended assessing the severity of symptoms over the past week [14].

## Assessments

The sociodemographic and medical characteristics of all participants were collected. For each patient, the DAS was completed by a psychiatrist. The following scales were also administered by the psychiatrist: the 17-item version of the HDRS [20] and the 18-item BPRS [23] from which were extracted, respectively, the six and five specific items to construct the PDAS, the Scale for the Assessment of Positive Symptoms (SAPS) to classify the type of delusion [26], the Mini Mental State Examination (MMSE) [27], the delusional rating item of the Schedule for Affective Disorders and Schizophrenia [19], and the CGI-S (Clinical Global Impression of severity) [28].

All data are available at Mendeley [29].

## Statistical analyses

Statistical analyses were performed with Stata software (version 13, StataCorp, College Station, TX). P-values <0.05 were considered to be statistically significant.

In accordance with recommendations set out by COSMIN [30] and Terwee et al. [31], we included a minimum number of 100 subjects to ensure satisfactory internal consistency evaluation and a sample size of at least 50 patients to guarantee an acceptable estimate of reliability.

In addition to descriptive statistics, the following psychometric properties of the DAS and PDAS scales were evaluated.

1. Scalability was assessed by Mokken analysis, generating a Loevinger coefficient of homogeneity for the PDAS total score. In Mokken analysis, a Loevinger coefficient of homogeneity ≥0.40 is indicative of scalability [32]. For the present analysis the BPRS scores were converted as follows: 1 = 0, 2–3 = 1, 4–5 = 2, 6 = 3, and 7 = 4.

2. Descriptive statistics and score distributions. In addition to mean and standard-deviation, score range, closeness of mean to median, and floor and ceiling effects [31] were calculated.

3. An exploratory factorial analysis (principal components analysis with varimax rotation) was performed to determine the scale structure. The number of factors was chosen according to usual recommendations: Kaiser criteria, plot of eigenvalues, and part of variance expressed by principal components. For items loading on several factors, the decision to do factor attribution was made according to psychometric results and clinical relevance.

4. Internal consistency was determined through Cronbach's alpha coefficient (minimum accepted value: 0.70 [33]), the item homogeneity coefficient (criterion value: ≥ 0.30 [34]) and the item-total correlation corrected for overlap (criterion value: ≥ 0.30 [35]).

5. Internal validity was determined by study of correlation between the domains making up the scale (standard, $p$ = 0.30–0.70 [36]).

6. Reproducibility. The intraclass correlation coefficient was used to determine inter-judge reliability. Values of concordance correlation coefficient greater than 0.70 were considered to be satisfactory [31].

7. Convergent validity. The relationships between DAS and PDAS scale scores and other quantitative measurements (HDRS, BPRS, CGI and SAPS) were studied with correlation coefficients (Pearson or Spearman according to statistical distribution). Association was

considered to be weak for a correlation coefficient between 0.3 and 0.5, and moderate to high for a correlation coefficient greater than 0.5.

## Results

Of the 100 patients initially recruited, 6 were excluded following reassessment of their diagnosis (3 bipolar disorders, 1 psychotic disorder, 1 dementia, and 1 depression caused by somatic disorders). The characteristics of the remaining 94 patients are given in Table 1. Fifty-one (55%) of the patients were female; they had a mean age of 62.2 ± 17.7 years old and 50% were

**Table 1. Participant characteristics.**

|  | Patients (n = 94) |
|---|---|
| Sex, female | 51 (55) |
| Age | 62.2 ± 17.7 |
| 65 years and more | 47 (50) |
| Education |  |
| Lower than high school | 43 (47) |
| High school diploma | 31 (34) |
| Higher than high school | 18 (20) |
| Number of Depressive Disorders |  |
| 1 (first episode) | 22 (23) |
| 2 | 38 (40) |
| 3 or more | 34 (37) |
| Treatment |  |
| Antidepressant | 88 (94) |
| Neuroleptic | 75 (80) |
| Anxiolytic | 81 (86) |
| Hypnotic | 27 (29) |
| Suicide |  |
| No | 18 (27) |
| Suicide ideations | 27 (41) |
| Suicide attempt | 21 (32) |
| Other Medical Comorbidities | 68 (72) |
| CGI |  |
| Mildly ill | 1 (1) |
| Moderately ill | 5 (5) |
| Markedly ill | 35 (37) |
| Severely ill | 45 (48) |
| Among the most extremely ill patients | 8 (9) |
| HDRS | 25.5 [24–30] |
| BPRS | 58 [51–66] |
| SAPS |  |
| Hallucinations | 1 [0–4] |
| Delusions | 4 [3–4] |
| Bizarre behaviour | 0 [0–1] |
| Positive formal thought disorder | 1 [0–2] |
| MMSE | 28 [26–29] |

Data are presented as mean ± SD or median [interquartile] and numbers (percentage).

more than 65 years old. Median HDRS and BPRS scores were 25.5 [24–30] and 58 [51–66], respectively. All patients had at least one clearly defined delusion and a score of ≥3 on the delusion severity rating item of the SADS.

## Scalability of the DAS and PDAS scales

The Loevinger coefficients of homogeneity derived from the Mokken analyses were 0.39 for DAS and 0.28 for PDAS. The predefined threshold for scalability (≥0.40) was not achieved despite significant results on both scales (p < 0.001). The Loevinger coefficients of homogeneity were 0.39 for HAM-D6 and 0.33 for BPRS5, both composing the PDAS.

## Factorial analysis

The 11-item and 15-item structures of the PDAS and DAS, respectively, were tested by factorial analysis with varimax rotation. Five factors accounting for 70.2% of the total variance were extracted and gave the best factor structure solution for PDAS (Table 2). HDRS items loaded on two factors and BPRS items on three. Only the item "Work and activities" (HDRS 7) loaded on two factors (1 and 3, both composed of HDRS items) and was finally associated with factor 3 designated "Fatigue/activities". Other dimensions were "Depression", "Emotional withdrawal/Blunted affect", "Psychotic symptoms" and "Suspiciousness".

For the DAS, four factors accounting for 60.7% of the total variance were extracted (Table 3). Factor 1 grouped together the "Conviction" and "Impact" dimensions of the original DAS and was named "Delusional conviction". Factor 2 grouped together the "Disorganization" and "Mood congruence" dimensions and item 8 "People/objects involved" of the original DAS. This factor formed the "Disorganization/Mood congruence" dimension. The "Bizarreness" dimension was retained and constituted factor 3. Two items, one negative and one positive, were loaded on factor 4: "Acting irrationally distrustful during the interview" (negative) and "Places/situations involved" (positive). This gave two dimensions "Acting irrationally during interview" and "Places/situations involved".

**Table 2. Factor loadings from the factor analysis on the PDAS scale.**

| Items | Dimensions | Factor 1 | Factor 2 | Factor 3 | Factor 4 | Factor 5 |
|---|---|---|---|---|---|---|
| | | Depression | Emotional withdrawal/ Blunted affect | Fatigue/activities | Psychotic symptoms | Suspiciousness |
| | **Variance explained 70.2%** | **18.1%** | **17.3%** | **13.0%** | **11.8%** | **10.0%** |
| HDRS 1 | Depressed mood | **0.7492** | 0.0172 | -0.2714 | 0.0747 | 0.2156 |
| HDRS 2 | Feelings of guilt | **0.6798** | -0.0316 | -0.2220 | -0.1408 | -0.2905 |
| HDRS 7 | Work and activities | **0.4365** | 0.3869 | **0.4578** | 0.0205 | 0.1132 |
| HDRS 8 | Retardation | **0.5819** | 0.2831 | 0.2317 | 0.0217 | 0.0696 |
| HDRS 10 | Psychic anxiety | **0.6173** | -0.2789 | 0.3525 | 0.1691 | -0.1213 |
| HDRS 13 | General somatic symptoms | -0.1275 | -0.1132 | **0.8598** | -0.0474 | 0.0234 |
| BPRS 3 | Emotional withdrawal | 0.1389 | **0.8645** | -0.0053 | 0.1295 | 0.1372 |
| BPRS 11 | Suspiciousness | 0.0031 | 0.0065 | 0.0149 | -0.0112 | **0.9512** |
| BPRS 12 | Hallucinatory behaviour | 0.0060 | 0.0240 | -0.1943 | **0.8563** | 0.0245 |
| BPRS 15 | Unusual thought content | 0.0534 | 0.1522 | 0.3654 | **0.7007** | -0.0612 |
| BPRS 16 | Blunted affect | -0.1310 | **0.9024** | -0.0765 | -0.0060 | -0.1109 |

All items that loaded higher than or equal to 0.40 are in bold.

**Table 3. Factor loadings from the factor analysis on the DAS scale.**

| Original DAS | Items | Dimensions | Factor 1 | Factor 2 | Factor 3 | Factor 4 |
|---|---|---|---|---|---|---|
| | | | Delusional conviction | Disorganization/Mood congruence | Bizarreness | Acting irrationally during interview, Places/situations involved |
| | | **Variance explained 60.7%** | **20.0%** | **19.5%** | **13.2%** | **8.2%** |
| Conviction | DAS 1 | Subjective feeling of certainty | **0.6920** | 0.1980 | 0.1629 | 0.0809 |
| | DAS 2 | Accommodation | **0.6248** | 0.2304 | 0.2244 | 0.0567 |
| Impact | DAS 3 | Acting on the belief | **0.5249** | 0.0803 | 0.2244 | 0.2932 |
| | DAS 4 | Acting irrationally distrustful during the interview | 0.3386 | 0.2394 | 0.1186 | **-0.6510** |
| | DAS 5 | Temporal pressure | **0.7776** | 0.0996 | 0.1956 | 0.0258 |
| | DAS 6 | Temporal pressure during interview | **0.6720** | 0.0255 | 0.1044 | -0.1648 |
| | DAS 7 | Emotional pressure | **0.6503** | -0.2698 | -0.0649 | 0.0234 |
| Extension | DAS 8 | People/objects involved | 0.3014 | **0.6684** | -0.0027 | 0.3406 |
| | DAS 9 | Places/situations involved | 0.2299 | 0.1998 | 0.2466 | **0.7135** |
| Bizarreness | DAS 10 | Implausibility/bizarreness | 0.1464 | 0.1674 | **0.8107** | 0.0990 |
| | DAS 11 | Relationship to cultural context | 0.1269 | 0.0526 | **0.8821** | 0.0129 |
| Disorganization | DAS 12 | Internal consistency | 0.0263 | **0.7458** | 0.3160 | 0.0449 |
| | DAS 13 | Cognitive integration | 0.0724 | **0.7291** | 0.4125 | -0.0181 |
| | DAS 14 | Temporal continuity | 0.1175 | **0.8068** | -0.1062 | -0.0194 |
| Mood congruence | DAS 15 | Mood congruence | -0.2347 | **0.6508** | 0.0588 | -0.1938 |

All items that loaded higher than or equal to 0.40 are in bold.

## Descriptive statistics and score distributions

The descriptive statistics and score distributions for the PDAS and DAS scales are given in Table 4. No ceiling effect was found for the PDAS. However, a floor effect was observed for the "Psychotic symptoms" and "Suspiciousness" dimensions.

A ceiling effect was observed for the "Places/situations involved" dimension of the DAS and floor effects for the dimensions of "Disorganization/Mood congruence", "Bizarreness", "Acting irrationally during interview" and "Places/situations involved".

## Internal consistency

The DAS showed good internal consistency with Cronbach's α greater than the minimum required coefficient of 0.70 for all dimensions (Table 5). The PDAS showed lower Cronbach's α (0.29 to 0.59). Only the "Emotional withdrawal/ Blunted affect" dimension had a Cronbach's α coefficient greater than 0.70.

## Inter-item correlations

The PDAS showed weak inter-item correlations (from 0.00 à 0.68). Correlations were only good for items belonging to the same dimension. The DAS showed stronger and more significant inter-item correlations (from 0.00 to 0.70) (S1 and S2 Tables).

**Table 4. Descriptive statistics and score distributions of the PDAS and DAS scales.**

| | Mean (SD) | Range | Median [IQR] | Floor effect (%) | Ceiling effect (%) |
|---|---|---|---|---|---|
| PDAS | 21.8 (4.9) | 10.3–31.3 | 21.7 [18.7–25.7] | 0.0 | 0.0 |
| Depression | 9.3 (2.6) | 4–15 | 9.0 [7.0–11.0] | 0.0 | 0.0 |
| Emotional withdrawal/ Blunted affect | 3.0 (1.9) | 0–8 | 2.7 [1.3–4.7] | 8.5 | 1.1 |
| Fatigue/activities | 4.9 (1.8) | 1–8 | 5.0 [4.0–6.0] | 0.0 | 11.7 |
| Psychotic symptoms | 2.9 (2.1) | 0–7.3 | 2.7 [1.3–4.7] | 16.0 | 1.1 |
| Suspiciousness | 1.7 (1.4) | 0–4 | 1.3 [0.7–2.7] | 23.4 | 6.4 |
| DAS | 28.9 (5.1) | 19–39 | 28 [25–33] | 0.0 | 0.0 |
| Delusional conviction | 14.2 (2.5) | 9–18 | 14 [13–16] | 0.0 | 6.4 |
| Disorganization/ Mood congruence | 7.8 (2.6) | 5–15 | 7 [6–10] | 22.3 | 2.1 |
| Bizarreness | 3.3 (1.2) | 2–6 | 3 [2–4] | 29.8 | 9.6 |
| Acting irrationally during interview | 1.4 (0.5) | 1–3 | 1 [1–2] | 62.8 | 1.1 |
| Places/situations involved | 2.2 (0.8) | 1–3 | 2 [2–3] | 21.3 | 42.6 |

SD: Standard Deviation; IQR: InterQuartile Range

## Item-total correlations

The PDAS showed moderate corrected item-total correlations with a coefficient greater than 0.30 for only three items, whereas the DAS showed good corrected item-total correlations (only three items were lower than 0.30) (S1 and S2 Tables).

## Inter-scale correlations

No correlation was observed between the dimensions of the PDAS (Fig 1 and S3 Table). The DAS dimensions had multiple correlations with only the "Acting irrationally during interview" dimension not correlating with any other.

## Inter-judge reliability

Twenty-six participants accepted to be assessed by a second psychiatrist. The PDAS and DAS both had good inter-judge reliability. Only one dimension showed an intraclass correlation coefficient lower than 0.70 in each scale: "Fatigue/activities" in PDAS (0.68 [0.46–0.89]) and

**Table 5. Cronbach's α and inter-judge reliability for the PDAS and DAS.**

| | Cronbach's α | ICC [95% CI] * |
|---|---|---|
| PDAS | 0.52 | 0.78 [0.62–0.94] |
| Depression | 0.59 | 0.83 [0.70–0.95] |
| Emotional withdrawal/ Blunted affect | 0.82 | 0.77 [0.61–0.93] |
| Fatigue/activities | 0.29 | 0.68 [0.46–0.89] |
| Psychotic symptoms | 0.44 | 0.93 [0.87–0.98] |
| Suspiciousness | - | 0.77 [0.61–0.93] |
| DAS | 0.81 | 0.88 [0.79–0.97] |
| Delusional conviction | 0.78 | 0.85 [0.75–0.96] |
| Disorganization/ Mood congruence | 0.80 | 0.94 [0.90–0.99] |
| Bizarreness | 0.73 | 0.49 [0.19–0.79] |
| Acting irrationally during interview | - | 1.00 |
| Places/situations involved | - | 0.77 [0.60–0.93] |

*ICC: Intraclass Correlation Coefficient and 95% CI: 95% Confidence Intervals, n = 26

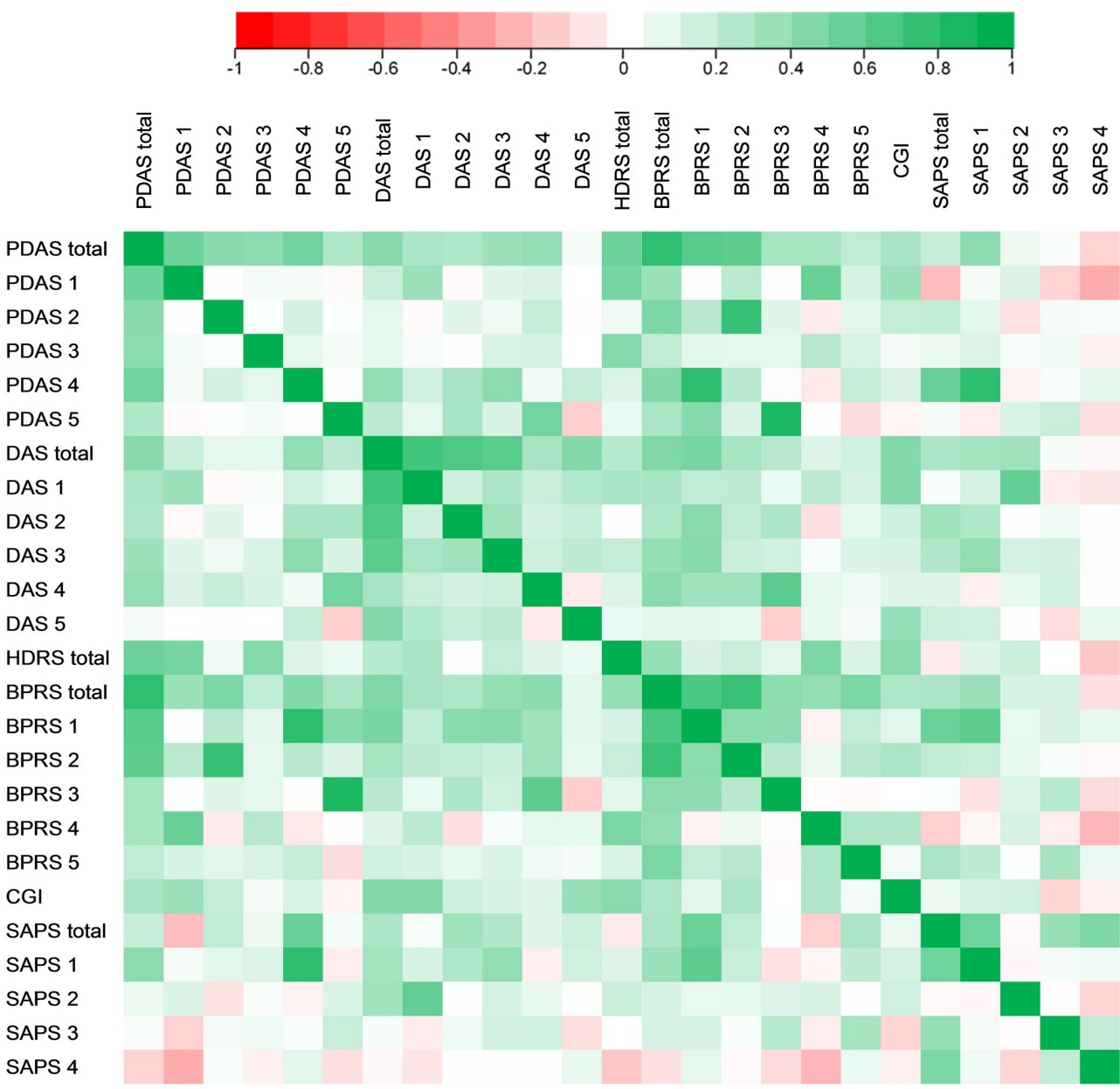

**Fig 1. Convergent validity for PDAS and DAS scales.** Correlations represented on a HEATMAP: in red, negative correlations and in green, positive correlations. A graduation of the color indicates the strength of the correlation. PDAS 1 = Depression; PDAS 2 = Emotional withdrawal/ Blunted affect; PDAS 3 = Fatigue/activities; PDAS 4 = Psychotic symptoms; PDAS 5 = Suspiciousness; DAS 1 = Delusional conviction; DAS 2 = Disorganization/ Mood congruence; DAS 3 = Bizarreness; DAS 4 = Acting irrationally during interview; DAS 5 = Places/situations involved; BPRS 1 = BPRS delusions hallucinations; BPRS 2 = BPRS hebephrenic; BPRS 3 = BPRS paranoia; BPRS 4 = BPRS melancholia anxious; BPRS 5 = BPRS acute psychotic; SAPS 1 = SAPS hallucinations; SAPS 2 = SAPS delusions; SAPS 3 = SAPS bizarre behavior; SAPS 4 = SAPS positive formal thought disorder.

"Bizarreness" in DAS (0.49 [0.19–0.79]). Others varied from 0.77 to 0.93 for PDAS and from 0.77 to 1.00 for DAS.

## Convergent validity

PDAS correlated with all the other scales tested (HDRS, BPRS, CGI and SAPS) and with several subscales (Fig 1 and S3 Table). The strongest correlation was with the total BPRS (r = 0.82,

p<0.0001). All PDAS dimensions correlated with the total BPRS, with correlation coefficients varying from 0.25 to 0.51. Only the two dimensions of "Depression" and "Fatigue/activities" from the HDRS correlated with the total HDRS. Certain PDAS dimensions were also associated with BPRS subscales, CGI and SAPS.

The DAS and all its dimensions except "Places/situations involved" correlated with the total BPRS (coefficient correlations from 0.31 to 0.51) and numerous BPRS dimensions. DAS was also associated with HDRS (r = 0.29, p = 0.05). DAS and all its dimensions were also associated with CGI and SAPS.

## Discussion

The present study describes the first cross-cultural evaluation of the French versions of the DAS and PDAS, including the first factor analysis of the PDAS, and an assessment of the floor and ceiling effects of the DAS and PDAS, which was not performed in the original studies.

Our study population was comparable to those in other published reports with regard to most sociodemographic and medical characteristics [4, 15, 37–39]. Medical treatment of the patients was consistent with guidelines on pharmacotherapy in MDDPsy [7–11, 40]. The DAS, PDAS, HDRS-17 and BPRS-18 total scores were similar to those observed in studies validating the DAS and PDAS scales [15, 41, 42]. The participants in this multicenter study were recruited from both university and non-university hospitals and assessed by trained psychiatrists.

### DAS

The French version of the DAS comprises five dimensions that differ from those of the original version. No item overlapped between the five factors. The first factor included six of the scale items, received the greatest loading, and accounted for 20% of variance. In comparison to the original DAS, the French version groups together the "Conviction" and "Impact" dimensions (except item 4). This is related to the observation of Meyers that there were some overlaps between the items of the "Conviction" and "Impact" dimensions from the original version of the DAS indicating a relationship between the constructs [15]. We decided to name this factor "Delusional conviction" to reflect the strength of the belief and of its emotional and behavioral impact. Factor 3 retains the two items of the "Bizarreness" factor from the original version. Factor 2 from the French version, "Disorganization/Mood congruence", comprises five items and accounts for 19.5% of variance. Compared to the original version, it includes item 8, "People/objects involved", and item 15, "Mood congruence" while factors 4 and 5 are each composed of a single item. Factor 4 from the DAS is made up solely of item 4, which assesses whether the patient is distrustful, and factor 5 of the PDAS is made up solely of BPRS item 11, which assesses the suspiciousness of the patient. These two dimensions are fairly independent of the other dimensions that make up the scales to which they belong and are well correlated with each other.

We applied the value of 15% for floor and ceiling effects [31]. The floor effect was observed for certain subscales, due partly perhaps to the absence of certain symptoms in our patients and partly to the construction of the scales: each item was rated from 1 to 3 with factor 3 comprising two items, and factors 4 and 5 one item each. This limits the response options even when responses are well distributed with items rated from 1 to 3.

The DAS had good internal consistency. The results obtained with the French version of the DAS were even better than those of the original version [15]. The internal consistency of the DAS scale as measured by the Cronbach α coefficient (0.81) was considered to be satisfactory and evidence of the good homogeneity of the scale. The Cronbach α coefficient scores of the three dimensions of the French version with enough items for the calculation (0.73, 0.78

and 0.80) were also considered acceptable. By comparison, the original DAS had an overall coefficient of 0.72 with a highest score of 0.85 for the Conviction dimension.

The DAS showed significant inter-item correlations, totaling 30/105 (r>0.30). We observed a satisfactory correlation (>0.60) between items 1 and 2 of dimension 1 of the French version of the DAS, the two items that make up the Conviction dimension of the original version and which were used as inclusion criteria in the STOP-PD therapeutic trial [16, 38, 43–46]. We also observed a satisfactory inter-item correlation (from 0.40 to 0.60) between items 10 and 11, which make up the Bizarreness dimension of the original and French versions of the DAS.

The DAS showed good item-total consistency except for item 15, "Mood congruence", which forms a single dimension in the original version of the DAS and was added to Kendler's dimensions of delusional experience, which provided the basis of the DAS [15].

Positive relationships were found between four of the five DAS dimensions: only factor 4, composed of the single item "Acting irrationally distrustful during interview", did not correlate with any other. We cannot make a comparison with the original version of the DAS because, to the best of our knowledge, no such analysis has been published.

The DAS had good inter-judge reliability, with good scores for each of these dimensions. The intraclass correlation coefficient (ICC) was 0.85 for dimension 1, which groups together the dimensions of Conviction and Impact from the original DAS, which in the study of Meyers, had the best ICC (respectively, 0.74 et 0.77). The ICC for dimension 2 was 0.94, which is an excellent score. This dimension groups together Disorganization (ICC of 0.37 in the study of Meyers), Mood Congruence (ICC of 0.51 in the study of Meyers) and one of the two items in Extension (ICC of 0.53 in the study of Meyers). According to Meyers, the reason for the weaker results for the dimensions of Disorganization and Extension was that "the raters have greater difficulty reliably assessing items that rate how delusions may change over time". He therefore recommended that "future investigators limit the timeframe for the domains of Disorganization and Extension to the week preceding the interview" [15]. Our study was performed in accordance with this recommendation, which could explain the improved ICC score for dimension 2. Dimension 3, which corresponds to the "Bizarreness" factor of the original DAS, was the exception with an ICC score of 0.49, which is nevertheless acceptable, and higher than that of 0.46 recorded in the study of Meyers. It is evidence, as Meyers clearly explained in his study, of the difficulty in assessing this dimension in patients with MDDPsy. The construction of this dimension of the DAS derives from Kendler's scale of dimensions of delusional experience, which was designed for patients with schizophrenia. Rating bizarreness can be more problematic in patients with MDDPsy and influenced by the investigator's experience and subjective assessment.

The DAS and all its dimensions except "Places/situations involved" correlated with the total BPRS, with coefficient correlations ranging from 0.31 to 0.51, and with numerous BPRS dimensions. The result for the total DAS was greater than that in the study of Meyers (0.50 as against 0.4) and four of the five dimensions of the French version were significantly correlated with the total score of the BPRS, as against two dimensions in the study of Meyers [15]. The French version of the DAS therefore fully accounted for psychotic symptoms. DAS was also associated with HDRS (r = 0.29, p = 0.005), with a weaker but nevertheless acceptable coefficient.

## PDAS

Surprisingly, despite a highly significant result, Mokken analysis did not confirm the scalability of the PDAS (Loevinger coefficient < 0.40). During validation of the semi-structured PDAS interview, Köse Çinar and Østergaard [41] also failed to demonstrate the scalability of the PDAS (Loevinger coefficient 0.17) at baseline. They stated that this could have been due to low

variability in the PDAS scores (22.8 ± 5.7). Our data showed a score with low variability (21.8 ± 4.9) close to that in their study [41] whereas the original PDAS validation study showed a higher score and variability (29.5 ± 11.3) [47]. These differences in PDAS score and variability could be due to the population studied. Despite being subject to the same inclusion criteria as in the article of Østergaard [16], our participants had a lower HDRS score, more hallucinations, more suicide attempts and all were inpatients. Another explanation could be the extraction of PDAS items from complete HDRS-17 and BPRS-18. In our study the specific semi-structured interview for the PDAS was not used, which could have affected the rating of the scale. In general, studies assessing the scalability of the PDAS demonstrate its scalability [16, 42, 47] and even Köse Çinar and Østergaard found a Loevinger coefficient > 0.40 at endpoint of their study [41].

With regard to the PDAS, the six items from the HDRS were divided into two dimensions, 1 and 3, which accounted, respectively, for 18.1 and 13.0% of variance while the five items from the BPRS were divided into three dimensions, 2, 4 and 5, which accounted, respectively, for 17.3, 11.8 and 10.0% of variance. Factor 1, which received the greatest loading, is made up of all the items from the "Depression" score of the HDRS. Item HDRS 7 "Work and activities" overlapped between the two factors that make up the HDRS subscale of the PDAS. We decided to place this item in dimension 3 because the psychometric qualities of the scale are better in this configuration.

The moderate results of all internal validity and consistency tests confirmed the construct of the PDAS scale and its scalability.

The PDAS had good inter-judge reliability. Only the "Fatigue/activities" dimension of the PDAS showed an intraclass correlation coefficient very slightly lower than 0.70. The reason could be that, as in the STOP-PD trial, half of our study population were elderly, at an age when they often experience fatigue of various causes, not necessarily imputable to MDDPsy and therefore difficult to assess. Likewise, it is difficult to gauge the impact of MDDPsy on changes in activity of subjects whose habits and behavior are affected by retirement and other age-related factors.

The PDAS correlated with all other scales tested (HDRS, BPRS, CGI and SAPS) and several subscales. The French version of the PDAS reflects the severity of the disease and accounts for both depressive and psychotic symptoms. The strongest correlation was with the total BPRS (r = 0.82, p<0.0001). All the PDAS dimensions correlated with the total BPRS with correlation coefficients varying from 0.25 to 0.51 due to the fact that 5 of the 11 items of the PDAS, i.e. three of the five dimensions, came from the BPRS. Only the two dimensions, "Depression" and "Fatigue/activities", from the HDRS correlated with the total HDRS, due to the fact that 6 of the 11 items of the PDAS, divided over two dimensions, derived from the HDRS.

One limitation of the present study is its failure to confirm the scalability of the PDAS scale. However, the results of internal validity and consistency tests tend towards confirmation. In addition, the good convergent validity and inter-judge reliability allow the scale to be used. However, inter-judge reliability was calculated from the assessments of only 26 patients, who had consented to a second interview. This relatively small number of volunteers can be explained by the symptoms of MDDPsy.

In conclusion, our study provides evidence of the good psychometric properties of the French versions of the DAS and PDAS. However, other studies are needed to confirm the scalability of the PDAS and the generalisability of the results to a larger population. This should help clinicians and researchers to assess cases of MDDPsy, which is a severe disorder, whose prognosis is marked by the risk of suicide and the lesser likelihood of patients achieving treatment response. The cross-cultural adaptation of this specific instrument should enable clinicians as of now to better assess the symptoms of MDDpsy, particularly psychotic features. It

should also enable investigators to propose French-speaking persons with MDDPsy as participants in international collaboration research projects. In both cases, the response to treatment and prognosis of MDDPsy should be improved.

## Supporting information

**S1 Table. Inter-item and corrected item-total correlations for PDAS.** Spearman correlation coefficients between HDRS and BPRS items making up the PDAS scale. * Corrected item-total correlations. Significant correlations (p<0.05) are in bold.
(DOCX)

**S2 Table. Inter-item and corrected item-total correlations for DAS.** Spearman correlation coefficients between HDRS and BPRS items making up the PDAS scale. * Corrected item-total correlations. Significant correlations (p<0.05) are in bold.
(DOCX)

**S3 Table. Convergent validity for PDAS and DAS scales.** Spearman correlation coefficients between PDAS, DAS, HDRS, BPRS, CGI and SAPS scales. Significant correlations (p<0.05) are in bold. PDAS 1 = Depression; PDAS 2 = Emotional withdrawal/ Blunted affect; PDAS 3 = Fatigue/activities; PDAS 4 = Psychotic symptoms; PDAS 5 = Suspiciousness; DAS 1 = Delusional conviction; DAS 2 = Disorganization/ Mood congruence; DAS 3 = Bizarreness; DAS 4 = Acting irrationally during interview; DAS 5 = Places/situations involved; BPRS 1 = BPRS delusions hallucinations; BPRS 2 = BPRS hebephrenic; BPRS 3 = BPRS paranoia; BPRS 4 = BPRS melancholia anxious; BPRS 5 = BPRS acute psychotic; SAPS 1 = SAPS hallucinations; SAPS 2 = SAPS delusions; SAPS 3 = SAPS bizarre behavior; SAPS 4 = SAPS positive formal thought disorder.
(DOCX)

**S1 File.**
(DOC)

## Acknowledgments

The authors thank Barnett S. Meyers for allowing them to validate the French version of the DAS scale and the membership of the French DAS/PDAS group: Aziz Amour, Esteban Arango, Florent Blanchard, Lucile Morel and Nadjim Zouaoui (Centre Hospitalier Spécialisé Sainte-Marie Le Puy-en-Velay), Dorothée Dhenain, Isaure Gaume and Silvia Alina Nechifor (Centre Hospitalier Spécialisé Sainte-Marie Clermont-Ferrand), Jonathan. Chabert, Isabelle Jalenques (leader of the group, ijalenques@chu-clermontferrand.fr), Sophie Lauron, Chloé Rachez, Fabien Rondepierre and Pierre-Antoine Thevenet (Centre Hospitalier Universitaire Clermont-Ferrand, Service de Psychiatrie de l'Adulte et Psychologie Médicale), Bruno Pereira (Centre Hospitalier Universitaire Clermont-Ferrand, Direction de la Recherche Clinique et de l'Innovation), Urbain Tauveron Jalenques and Jeffrey Watts (Université Clermont Auvergne).

## Author Contributions

**Conceptualization:** Isabelle Jalenques, Silvia Alina Nechifor, Fabien Rondepierre.

**Formal analysis:** Bruno Pereira, Fabien Rondepierre.

**Investigation:** Silvia Alina Nechifor, Lucile Morel, Florent Blanchard.

**Methodology:** Chloé Rachez, Urbain Tauveron Jalenques, Fabien Rondepierre.

**Project administration:** Fabien Rondepierre.

**Resources:** Chloé Rachez, Fabien Rondepierre.

**Software:** Bruno Pereira.

**Supervision:** Isabelle Jalenques, Fabien Rondepierre.

**Validation:** Isabelle Jalenques.

**Visualization:** Chloé Rachez, Sophie Lauron.

**Writing – original draft:** Isabelle Jalenques, Fabien Rondepierre.

**Writing – review & editing:** Isabelle Jalenques, Chloé Rachez, Sophie Lauron.

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
