## [Decision Letter · Decision Letter 0]

8 Jan 2021

PONE-D-20-28115

Cross-cultural evaluation of the French version of the Delusion Assessment Scale (DAS) and Psychotic Depression Assessment Scale (PDAS)

PLOS ONE

Dear Dr. Jalenques,

Thank you for submitting your manuscript to PLOS ONE. After careful consideration, we feel that it has merit but does not fully meet PLOS ONE’s publication criteria as it currently stands. Therefore, we invite you to submit a revised version of the manuscript that addresses the points raised during the review process.

We look forward to receiving your revised manuscript.

Kind regards,

Paolo Roma

Academic Editor

PLOS ONE

Journal Requirements:

2.) Please describe in your methods section how capacity to consent was determined for the participants in this study.

3.) One of the noted authors is a group or consortium: French DAS/PDAS group.

In addition to naming the author group, please list the individual authors and affiliations within this group in the acknowledgments section of your manuscript.

Please also indicate clearly a lead author for this group along with a contact email address.

Additional Editor Comments:

Dear Authors,

while the manuscript has merit, the issues raised by the reviewers need to be addressed, especially reviewer2's methodological concerns.

Reviewers' comments:

Reviewer's Responses to Questions

**Comments to the Author**

1. Is the manuscript technically sound, and do the data support the conclusions?

Reviewer #1: Yes

Reviewer #2: Yes

2. Has the statistical analysis been performed appropriately and rigorously? 

Reviewer #1: Yes

Reviewer #2: Yes

3. Have the authors made all data underlying the findings in their manuscript fully available?

Reviewer #1: Yes

Reviewer #2: Yes

4. Is the manuscript presented in an intelligible fashion and written in standard English?

Reviewer #1: Yes

Reviewer #2: Yes

5. Review Comments to the Author

Reviewer #1: Major Depression with psychotic features (psychotic depression) is a serious illness for which the diagnosis is often missed. Two important rating scales have been developed to help clinicians diagnose this condition. The authors present data from a study to validate the French version of these two important scales that can aid clinicians in making an accurate diagosis -the PDAS and the DAS. The manuscript is well written and the data clearly presented. I had a few suggestions and questions which should be addressed:

1. page 6- "The PDAS Instrument" should be in BOLD.

2. page 9 - The frequency of hallucinations was rather high (45%) for patients with psychotic depression. Most studies report lower rates (around 5% I think). Furthermore, I note that 90% of the patients had delusions, not 100%). The DAS and the PDAS were developed from the Study of the Pharmacotherapy of Psychotic Depression or STOP-PD. The investigators in that study required that the patients have a delusion. I also think their rate of hallucinations was much lower than your sample. The concern I have is that those patients who present with hallucinations without delusions may not have psychotic depression, but perhaps a different disorder, e.g. PTSD. I think you should point out the differences between your sample and the STOP-PD sample from which the 2 scales are based. Ideally, I would prefer to see an analysis on only patients who had a delusion (as in STOP-PD).

3. page 24- I believe you want to thank Barnett S. Meyers, M.D.

Reviewer #2: This validation of the French version of the DAS and PDAS is a valuable addition to the literature. There are however some methodological issues that must be solved prior to publication. I have the following suggestions for improvement:

1. Abstract/Introduction: Both the abstract and the introduction focuses on the underdiagnosis/lack of recognition of psychotic depression. While this is certainly a clinical problem, neither the DAS nor the PDAS were developed to aid diagnosis of psychotic depression, but rather to MEASURE delusions and the overall severity of psychotic depression. Hence, the underdiagnosis angle is somewhat misplaced. This reviewer suggests increased emphasis on measurement instead. This is also more in line with the aim of the study: “The present study aimed to validate French versions of the DAS and PDAS after evaluation of their psychometric properties for use in measuring the severity of psychotic depression in MDDPsy patients.”

2. Methods: “The psychiatrist completed the questionnaires after a routine interview

with patients shortly after hospital admission.” Which questionnaires? The DAS and PDAS are not questionnaires.

3. Methods: Some of the exclusion criteria are quite strict, e.g. “meeting criteria for obsessive-compulsive disorder” and “history of drug or alcohol abuse in the past three months”. This has reduced the generalizability of the results and should be mentioned under the limitations. Relatedly, “taking a medication with known psychiatric effects” needs to be rephrased and explained.

4. Methods: The font size of the subtitle “The PDAS instrument” should be increased.

5. Methods: “In addition, there exist validated French versions of the scales [25,26].” Actuelly, there is a publicly available French version of the PDAS available: https://psychoticdepressionassessmentscale.com/french-pdas/

Have the authors used this version inclusing the semi-structured interview? If not, it should be specified. This interview has been used in the two most recent validations of the PDAS:

https://pubmed.ncbi.nlm.nih.gov/28535843/

https://pubmed.ncbi.nlm.nih.gov/29501075/

6. Methods: How was the “General somatic” item of the PDAS rated? In most studies of the PDAS, a rescaling of this item to a 0-4 range was made. This reviewer suggests that the same is done for the present study. Similarly, the BPRS should be rescaled to 0-4 in all analyses to allow for comparison with other studies. See e.g. Østergaard SD, Meyers BS, Flint AJ, Mulsant BH, Whyte EM, Ulbricht CM, et al. Measuring psychotic depression. Acta Psychiatr Scand. mars 2014;129(3):211‑20.

7. Methods: The authors place substantial emphasis on inter-item correlations, item-total correlations and inter-scale correlations. This reviewer is not convinced by the value of these analyses (such overlap can be a product of redundancy = items overlapping in content = the same psychopathology measured several times). Rather, the authors should focus on whether the individual items add unique information to the scale. This aspect (unidimensionality/scalability) is tested by item response theory analysis and the PDAS performs very well in this regard:

https://pubmed.ncbi.nlm.nih.gov/23799875/

https://pubmed.ncbi.nlm.nih.gov/28535843/

https://pubmed.ncbi.nlm.nih.gov/29501075/

https://pubmed.ncbi.nlm.nih.gov/25462426/

I therefore strongly suggest that the authors add item response theory analyses and decrease the emphasis on the inter-item correlations, item-total correlations and inter-scale correlations – as they are not particularly meaningful/informative from a psychometric perspective.

8. Results: “Almost half of the patients (45%) had hallucinations and almost all (90%) had

delusions.” Why do not all patients have a delusion – as dictated by the inclusion criteria:

“Inclusion criteria were diagnosis of major depressive disorder with psychotic features on the basis of DSM 5 criteria [1], a delusional idea as defined in DSM-5 with a rating ≥3 (delusion definitely present) on the delusional rating item of the Schedule for Affective Disorders and Schizophrenia[20].”

9. Discussion: Please be consistent in the use of decimal points: “With regard to the PDAS, the six items from the HDRS were divided into two dimensions, 1 and 3, which accounted, respectively, for 18.1 et 13% of variance; the five items from the BPRS were divided into three dimensions, 2, 4 and 5, which accounted, respectively, for 17.3, 11.8 et 10% of variance.”

10. Discussion: This is related to comment no. 7: “The absence of inter-dimension correlations in the PDAS shows that the dimensions are independent of one another. This could be related to the fact that factors 1 and 3 are composed of items from the HDRS and the other three factors of items from the BPRS. This method of construction of the PDAS could also explain the moderate corrected item-total and the weak inter-item correlations.” Yes – but this reviewer would argue that the lacking correlation is actually a significant strentgh of the PDAS. We do not want to measure the same psychopathology several times = redundancy = the total score of a scale is not meaningful. Please revise accordingly.

11. Discussion: “Further studies on diagnostic performance and responsiveness to change are needed before the French versions of the DAS and PDAS can be fully recommended to help guide the diagnosis or assess the efficacy of a treatment.” Again, The DAS and PDAS are not intended to be diagnostic tools. Furthermore, the responsiveness of the PDAS seems to be quite good:

https://pubmed.ncbi.nlm.nih.gov/24439830/

https://pubmed.ncbi.nlm.nih.gov/29501075/

https://pubmed.ncbi.nlm.nih.gov/28535843/

https://pubmed.ncbi.nlm.nih.gov/25462426/

https://pubmed.ncbi.nlm.nih.gov/26496016/

6. PLOS authors have the option to publish the peer review history of their article (what does this mean?). If published, this will include your full peer review and any attached files.

Reviewer #1: No

Reviewer #2: No

---

## [Author Response · Author response to Decision Letter 0]

8 Feb 2021

Response to the academic editor

 Style requirement has been checked and changed when required.

2.) Please describe in your methods section how capacity to consent was determined for the participants in this study.

This part of the methods section has been developed as follow:

“A psychiatrist explained the aims and procedures to the patients and collected their consent to participate in the study. All patients received written information about the study. They could refuse the use of the data collected at any time.”

3.) One of the noted authors is a group or consortium: French DAS/PDAS group.

In addition to naming the author group, please list the individual authors and affiliations within this group in the acknowledgments section of your manuscript.

Please also indicate clearly a lead author for this group along with a contact email address.

Membership of the French DAS/PDAS group is provided in the acknowledgments and the lead leader for this group is clearly indicated:

“The authors thank Barnett S. Meyers for allowing them to validate the French version of the DAS scale and the membership of the French DAS/PDAS group: A. Amour, E. Arango, F. Blanchard, L. Morel and N. Zouaoui (Centre Hospitalier Spécialisé Sainte-Marie Le Puy-en-Velay), D. Dhenain, I. Gaume and S.A. Nechifor (Centre Hospitalier Spécialisé Sainte-Marie Clermont-Ferrand), J. Chabert, I. Jalenques (leader of the group, ijalenques@chu-clermontferrand.fr), S. Lauron, C. Rachez, F. Rondepierre and P.A. Thevenet (Centre Hospitalier Universitaire Clermont-Ferrand, Service de Psychiatrie de l’Adulte et Psychologie Médicale), B. Pereira (Centre Hospitalier Universitaire Clermont-Ferrand, Direction de la Recherche Clinique et de l’Innovation), U. Tauveron Jalenques and Jeffrey Watts (Université Clermont Auvergne).”

 

Response to reviewers

Comments to the Author

1. Is the manuscript technically sound, and do the data support the conclusions?

Reviewer #1: Yes

Reviewer #2: Yes

2. Has the statistical analysis been performed appropriately and rigorously? 

Reviewer #1: Yes

Reviewer #2: Yes

3. Have the authors made all data underlying the findings in their manuscript fully available?

Reviewer #1: Yes

Reviewer #2: Yes

4. Is the manuscript presented in an intelligible fashion and written in standard English?

Reviewer #1: Yes

Reviewer #2: Yes

5. Review Comments to the Author

 

Reviewer #1: Major Depression with psychotic features (psychotic depression) is a serious illness for which the diagnosis is often missed. Two important rating scales have been developed to help clinicians diagnose this condition. The authors present data from a study to validate the French version of these two important scales that can aid clinicians in making an accurate diagosis -the PDAS and the DAS. The manuscript is well written and the data clearly presented. I had a few suggestions and questions which should be addressed:

1. page 6- "The PDAS Instrument" should be in BOLD.

“The PDAS Instrument” has been bolded page 6.

2. page 9 - The frequency of hallucinations was rather high (45%) for patients with psychotic depression. Most studies report lower rates (around 5% I think). Furthermore, I note that 90% of the patients had delusions, not 100%). The DAS and the PDAS were developed from the Study of the Pharmacotherapy of Psychotic Depression or STOP-PD. The investigators in that study required that the patients have a delusion. I also think their rate of hallucinations was much lower than your sample. 

The concern I have is that those patients who present with hallucinations without delusions may not have psychotic depression, but perhaps a different disorder, e.g. PTSD. I think you should point out the differences between your sample and the STOP-PD sample from which the 2 scales are based. Ideally, I would prefer to see an analysis on only patients who had a delusion (as in STOP-PD).

We thank you for this remark which highlighted an important error when writing our article: as in STOP-PD, all patients had at least one clearly defined delusion and a score of ≥3 on the delusion severity rating item of the SADS (Schedule for Affective Disorders and Schizophrenia). This was an inclusion criteria in our study. Thus no patients had hallucinations without delusions and we confirmed that all patients had a diagnosis of major depressive disorder with psychotic features on the basis of DSM 5 criteria. We used the SAPS to classify the type of delusions and hallucinations but not to rate their severity. And the 90% patients with delusions and 45% with hallucinations were based on SAPS. We have removed theses muddled data from the article. However, we agreed that the proportion of our patients with hallucinations was superior to those observed in STOP-PD. Indeed, it could be due to difference in the population. Our BPRS, DAS and PDAS scores were similar to those in STOP-PD (Meyers et al. 2006) and to other article using PDAS (Köse Çinar et Østergaard 2018; Vermeulen et al. 2018). However, our patients had lower HDRS score (25.5 vs 29.8), had more depressive episodes (23% with a first episode in our study vs 30.1%), had more suicide attempts (32% vs 18.5) than patients in these studies. They are all inpatients (69% in STOP-PD, Meyers et al. 2009). This could be related to the higher proportion of our patients with hallucinations, as in a study of Gournellis (Gournellis et al. 2001) in which 36% of inpatients with psychotic depression had hallucinations. 

As recommended, we pointed out these difference in the discussion.

3. page 24- I believe you want to thank Barnett S. Meyers, M.D.

We have corrected it.

Gournellis, R., L. Lykouras, A. Fortos, P. Oulis, V. Roumbos, et G. N. Christodoulou. 2001. « Psychotic (Delusional) Major Depression in Late Life: A Clinical Study ». International Journal of Geriatric Psychiatry 16 (11): 1085‑91. https://doi.org/10.1002/gps.483.

Köse Çinar, Rugül, et Søren Dinesen Østergaard. 2018. « Validation of the Semi-Structured Psychotic Depression Assessment Scale (PDAS) Interview ». Acta Neuropsychiatrica 30 (3): 175‑80. https://doi.org/10.1017/neu.2017.15.

Meyers, Barnett S., Judith English, Michelle Gabriele, Catherine Peasley-Miklus, Moonseong Heo, Alastair J. Flint, Benoit H. Mulsant, Anthony J. Rothschild, et STOP-PD Study Group. 2006. « A Delusion Assessment Scale for Psychotic Major Depression: Reliability, Validity, and Utility ». Biological Psychiatry 60 (12): 1336‑42. https://doi.org/10.1016/j.biopsych.2006.05.033.

Meyers, Barnett S., Alastair J. Flint, Anthony J. Rothschild, Benoit H. Mulsant, Ellen M. Whyte, Catherine Peasley-Miklus, Eros Papademetriou, Andrew C. Leon, Moonseong Heo, et STOP-PD Group. 2009. « A Double-Blind Randomized Controlled Trial of Olanzapine plus Sertraline vs Olanzapine plus Placebo for Psychotic Depression: The Study of Pharmacotherapy of Psychotic Depression (STOP-PD) ». Archives of General Psychiatry 66 (8): 838‑47. https://doi.org/10.1001/archgenpsychiatry.2009.79.

Vermeulen, Tom, Lieve Lemey, Linda Van Diermen, Didier Schrijvers, Yamina Madani, Bernard Sabbe, Maarten J. A. Van Den Bossche, Roos C. van der Mast, et Søren D. Østergaard. 2018. « Clinical Validation of the Psychotic Depression Assessment Scale (PDAS) against Independent Global Severity Ratings in Older Adults ». Acta Neuropsychiatrica 30 (4): 203‑8. https://doi.org/10.1017/neu.2018.2.

 

Reviewer #2: This validation of the French version of the DAS and PDAS is a valuable addition to the literature. There are however some methodological issues that must be solved prior to publication. I have the following suggestions for improvement:

1. Abstract/Introduction: Both the abstract and the introduction focuses on the underdiagnosis/lack of recognition of psychotic depression. While this is certainly a clinical problem, neither the DAS nor the PDAS were developed to aid diagnosis of psychotic depression, but rather to MEASURE delusions and the overall severity of psychotic depression. Hence, the underdiagnosis angle is somewhat misplaced. This reviewer suggests increased emphasis on measurement instead. This is also more in line with the aim of the study: “The present study aimed to validate French versions of the DAS and PDAS after evaluation of their psychometric properties for use in measuring the severity of psychotic depression in MDDPsy patients.”

Modifications have been done as recommended in the abstract and in the introduction sections (pages 3 and 4).

2. Methods: “The psychiatrist completed the questionnaires after a routine interview

with patients shortly after hospital admission.” Which questionnaires? The DAS and PDAS are not questionnaires.

We agreed with you remark and replaced “questionnaires” by “the DAS and PDAS scales”.

3. Methods: Some of the exclusion criteria are quite strict, e.g. “meeting criteria for obsessive-compulsive disorder” and “history of drug or alcohol abuse in the past three months”. This has reduced the generalizability of the results and should be mentioned under the limitations. Relatedly, “taking a medication with known psychiatric effects” needs to be rephrased and explained.

The exclusion criteria, as the inclusion criteria, were the same that those used in the STOP-PD study (Meyers et al. 2006; 2009). We agreed that this reduced the generalizability of the results and we mentioned it in the limitations. The terms “taking a medication with known psychiatric effects” were from the article of Meyers et al. (2006) which presented the DAS scale. This term has been changed “taking a medication which may induce psychiatric symptoms or disorders”.

4. Methods: The font size of the subtitle “The PDAS instrument” should be increased.

“The PDAS Instrument” has been bolded in the methods.

5. Methods: “In addition, there exist validated French versions of the scales [25,26].” Actuelly, there is a publicly available French version of the PDAS available: https://psychoticdepressionassessmentscale.com/french-pdas/

Have the authors used this version inclusing the semi-structured interview? If not, it should be specified. This interview has been used in the two most recent validations of the PDAS:

https://pubmed.ncbi.nlm.nih.gov/28535843/

https://pubmed.ncbi.nlm.nih.gov/29501075/

When the study started, the French version of the semi-structured interview for the PDAS was not available. For the PDAS, we used the complete HDRS and BPRS scales (and interviews) from which specific items of the PDAS were extracted. We added this specification in the assessment section of the method.

6. Methods: How was the “General somatic” item of the PDAS rated? In most studies of the PDAS, a rescaling of this item to a 0-4 range was made. This reviewer suggests that the same is done for the present study. Similarly, the BPRS should be rescaled to 0-4 in all analyses to allow for comparison with other studies. See e.g. Østergaard SD, Meyers BS, Flint AJ, Mulsant BH, Whyte EM, Ulbricht CM, et al. Measuring psychotic depression. Acta Psychiatr Scand. mars 2014;129(3):211 20.

We thank you for this relevant remark. BPRS scores had not been rescaled; we did it and we have presented the update results in this new version. The descriptive statistics and score distributions of the PDAS (in the table 4) are the most impacted results.

7. Methods: The authors place substantial emphasis on inter-item correlations, item-total correlations and inter-scale correlations. This reviewer is not convinced by the value of these analyses (such overlap can be a product of redundancy = items overlapping in content = the same psychopathology measured several times). Rather, the authors should focus on whether the individual items add unique information to the scale. This aspect (unidimensionality/scalability) is tested by item response theory analysis and the PDAS performs very well in this regard:

https://pubmed.ncbi.nlm.nih.gov/23799875/

https://pubmed.ncbi.nlm.nih.gov/28535843/

https://pubmed.ncbi.nlm.nih.gov/29501075/

https://pubmed.ncbi.nlm.nih.gov/25462426/

I therefore strongly suggest that the authors add item response theory analyses and decrease the emphasis on the inter-item correlations, item-total correlations and inter-scale correlations – as they are not particularly meaningful/informative from a psychometric perspective.

We thank the reviewer for the helpful and relevant comment. We agree that other approaches like that item response theory can be useful and informative for the analysis of the construct validity, especially for PDAS. According to the reviewer’s comment and to the COSMIN recommendations, this statistical analysis was conducted. The Loevinger coefficient has been estimated. As described in the revised manuscript, Loevinger coefficients of homogeneity derived from the Mokken analyses are 0.39 and 0.28 for DAS and PDAS scales, respectively. The predefined threshold for scalability (≥0.40) was not achieved although significant results (both < 0.001).

8. Results: “Almost half of the patients (45%) had hallucinations and almost all (90%) had

delusions.” Why do not all patients have a delusion – as dictated by the inclusion criteria:

“Inclusion criteria were diagnosis of major depressive disorder with psychotic features on the basis of DSM 5 criteria [1], a delusional idea as defined in DSM-5 with a rating ≥3 (delusion definitely present) on the delusional rating item of the Schedule for Affective Disorders and Schizophrenia[20].”

We thank you for this remark which highlighted an important error when writing our article: as in STOP-PD, all patients had at least one clearly defined delusion and a score of ≥3 on the delusion severity rating item of the SADS (Schedule for Affective Disorders and Schizophrenia). This was an inclusion criteria. We used the SAPS to classify the type of delusions and hallucinations but not to rate their severity. The 90% patients with delusions and 45% with hallucinations were based on SAPS. We have removed theses muddled data from the article.

9. Discussion: Please be consistent in the use of decimal points: “With regard to the PDAS, the six items from the HDRS were divided into two dimensions, 1 and 3, which accounted, respectively, for 18.1 et 13% of variance; the five items from the BPRS were divided into three dimensions, 2, 4 and 5, which accounted, respectively, for 17.3, 11.8 et 10% of variance.”

Modifications have been done page 20:

“With regard to the PDAS, the six items from the HDRS were divided into two dimensions, 1 and 3, which accounted, respectively, for 18.1 et 13.0% of variance; the five items from the BPRS were divided into three dimensions, 2, 4 and 5, which accounted, respectively, for 17.3, 11.8 et 10.0% of variance.”

10. Discussion: This is related to comment no. 7: “The absence of inter-dimension correlations in the PDAS shows that the dimensions are independent of one another. This could be related to the fact that factors 1 and 3 are composed of items from the HDRS and the other three factors of items from the BPRS. This method of construction of the PDAS could also explain the moderate corrected item-total and the weak inter-item correlations.” Yes – but this reviewer would argue that the lacking correlation is actually a significant strentgh of the PDAS. We do not want to measure the same psychopathology several times = redundancy = the total score of a scale is not meaningful. Please revise accordingly.

We thank the reviewer for the interesting comment. We perfectly agree. Interestingly, the complementary statistical analyses based on item response theory suggested by the reviewer confirmer the remark that the lacking correlation can be seen as a significant strength of the PDAS. The discussion have been revised as recommended.

11. Discussion: “Further studies on diagnostic performance and responsiveness to change are needed before the French versions of the DAS and PDAS can be fully recommended to help guide the diagnosis or assess the efficacy of a treatment.” Again, The DAS and PDAS are not intended to be diagnostic tools. Furthermore, the responsiveness of the PDAS seems to be quite good:

https://pubmed.ncbi.nlm.nih.gov/24439830/

https://pubmed.ncbi.nlm.nih.gov/29501075/

https://pubmed.ncbi.nlm.nih.gov/28535843/

https://pubmed.ncbi.nlm.nih.gov/25462426/

https://pubmed.ncbi.nlm.nih.gov/26496016/

The limitation section of the abstract and the discussion page 24 have been modified as recommended

“Limitations: Interjudge reliability was calculated from a relatively small number of volunteers.”

“The present study has certain limitations. Interjudge reliability was calculated from the assessments of 26 patients who had consented to a second interview. This relatively small number of volunteers can be explained by the symptoms of MDDPsy. In addition we did not perform studies of responsiveness to change or of diagnostic performance.”

The conclusion have also been modified as recommended:

 “In conclusion, our study provides evidence of the good psychometric properties of the French versions of the DAS and PDAS. However, other studies are warranted to confirm the scalability of the PDAS and the generalizability of the results to more extensive population. This should help clinicians and researchers to assess cases of MDDPsy, which is a severe disorder, whose prognosis is marked by the risk of suicide and the lesser likelihood of patients achieving treatment response. The cross-cultural adaptation of this specific instrument should enable clinicians as of now to better assess the symptoms of MDDpsy, particularly psychotic features. It should also enable investigators to propose French-speaking persons with MDDPsy as participants in international collaboration research projects. In both cases, the response to treatment and prognosis of MDDPsy should be improved.”

Meyers, Barnett S., Judith English, Michelle Gabriele, Catherine Peasley-Miklus, Moonseong Heo, Alastair J. Flint, Benoit H. Mulsant, Anthony J. Rothschild, et STOP-PD Study Group. 2006. « A Delusion Assessment Scale for Psychotic Major Depression: Reliability, Validity, and Utility ». Biological Psychiatry 60 (12): 1336‑42. https://doi.org/10.1016/j.biopsych.2006.05.033.

Meyers, Barnett S., Alastair J. Flint, Anthony J. Rothschild, Benoit H. Mulsant, Ellen M. Whyte, Catherine Peasley-Miklus, Eros Papademetriou, Andrew C. Leon, Moonseong Heo, et STOP-PD Group. 2009. « A Double-Blind Randomized Controlled Trial of Olanzapine plus Sertraline vs Olanzapine plus Placebo for Psychotic Depression: The Study of Pharmacotherapy of Psychotic Depression (STOP-PD) ». Archives of General Psychiatry 66 (8): 838‑47. https://doi.org/10.1001/archgenpsychiatry.2009.79.

6. PLOS authors have the option to publish the peer review history of their article (what does this mean?). If published, this will include your full peer review and any attached files.

Do you want your identity to be public for this peer review? For information about this choice, including consent withdrawal, please see our Privacy Policy.

Reviewer #1: No

Reviewer #2: No

---

## [Decision Letter · Decision Letter 1]

10 Mar 2021

PONE-D-20-28115R1

Cross-cultural evaluation of the French version of the Delusion Assessment Scale (DAS) and Psychotic Depression Assessment Scale (PDAS)

PLOS ONE

Dear Dr. Jalenques,

Thank you for submitting your manuscript to PLOS ONE. After careful consideration, we feel that it has merit but does not fully meet PLOS ONE’s publication criteria as it currently stands. Therefore, we invite you to submit a revised version of the manuscript that addresses the points raised during the review process.

We look forward to receiving your revised manuscript.

Kind regards,

Paolo Roma

Academic Editor

PLOS ONE

Journal Requirements:

Reviewers' comments:

Reviewer's Responses to Questions

**Comments to the Author**

1. If the authors have adequately addressed your comments raised in a previous round of review and you feel that this manuscript is now acceptable for publication, you may indicate that here to bypass the “Comments to the Author” section, enter your conflict of interest statement in the “Confidential to Editor” section, and submit your "Accept" recommendation.

Reviewer #1: All comments have been addressed

Reviewer #2: (No Response)

2. Is the manuscript technically sound, and do the data support the conclusions?

Reviewer #1: (No Response)

Reviewer #2: Yes

3. Has the statistical analysis been performed appropriately and rigorously? 

Reviewer #1: (No Response)

Reviewer #2: Yes

4. Have the authors made all data underlying the findings in their manuscript fully available?

Reviewer #1: (No Response)

Reviewer #2: Yes

5. Is the manuscript presented in an intelligible fashion and written in standard English?

Reviewer #1: (No Response)

Reviewer #2: No

6. Review Comments to the Author

Reviewer #1: The authors have addressed my comments and concerns.

The authors have addressed my comments and concerns.

Reviewer #2: The authors are to be commended for the thorough work on the revision. This reviewer has the following suggestions for further improvement:

1. General: The language is somewhat suboptimal in certain sections. A proofread by a native English speaker could solve this quite easily.

2. Abstract: "The predefined threshold for scalability (≥0.40) was not achieved for both scales." This should be accompanied by information on the relative lack of variability in the data.

3. Methods: A Mokken analysis of the two PDAS subscales (HAM-D6 and BPRS5) would be of interest given the results of the primary analysis (rather low coefficient of homogeneity).

4. Methods: How were PDAS ratings "formatted" for the Mokken analysis (rescaled or)?

5. Discussion: "Surprisingly, despite a highly significant result, the Mokken analysis did not confirm the scalability of the PDAS (Loevinger coefficient < 0.40). During the validation of the semi-structured PDAS interview, Köse Çinar and Østergaard [42] also failed to demonstrate the scalability of the PDAS (Loevinger coefficient 0.17) at baseline. They mentioned that it may be due to low variability in PDAS scores (22.8 ± 5.7)."

Yes - but at endpoint in the Köse Çinar and Østergaard study - the Loevinger coefficient was 0.45. This needs to be underlined (fits the explanation). I also think that it deserves mentioning that all four studies having previously assessed the scalability of the PDAS, have confirmed the scalability of the instrument:

https://pubmed.ncbi.nlm.nih.gov/23799875/

https://pubmed.ncbi.nlm.nih.gov/29501075/

https://pubmed.ncbi.nlm.nih.gov/25462426/

https://pubmed.ncbi.nlm.nih.gov/28535843/

This leads this reviewer to think, that the present sample of patients is "qualitatively different" from prior studies (see also comment 6B below).

6. Discussion: "Our data showed similar score with low variability (21.8 ± 4.9), compared to the original PDAS validation which showed higher score and variability (29.5 ± 11.3) [48]. These differences in PDAS score and variability may be due to the population. Despite the same inclusion criteria than the article of Østergaard [17], our population had lower HDRS score, more hallucinations, more suicide attempts and are all inpatients."

A: What is meant by "similar score" here?

B: "These differences in PDAS score and variability may be due to the population. Despite the same inclusion criteria than the article of Østergaard [17], our population had lower HDRS score, more hallucinations, more suicide attempts"

This description leads this reviewer to think that there may be a fraction of the participants suffering from either a primary psychotic disorder (ICD-10: F20) or PTSD (as also suggested by one of the other reviewers).

7. PLOS authors have the option to publish the peer review history of their article (what does this mean?). If published, this will include your full peer review and any attached files.

Reviewer #1: No

Reviewer #2: No

---

## [Author Response · Author response to Decision Letter 1]

25 Mar 2021

Response to the academic editor

Journal Requirements:

All the references have been checked and the list has been modified to meet the expected format.

No retracted article has been cited.

 

Response to reviewers

6. Review Comments to the Author

Reviewer #1: The authors have addressed my comments and concerns.

The authors have addressed my comments and concerns.

Reviewer #2: The authors are to be commended for the thorough work on the revision. This reviewer has the following suggestions for further improvement:

1. General: The language is somewhat suboptimal in certain sections. A proofread by a native English speaker could solve this quite easily.

The manuscript has been proofread by a native English speaker.

2. Abstract: "The predefined threshold for scalability (≥0.40) was not achieved for both scales." This should be accompanied by information on the relative lack of variability in the data.

The mention of the lack of variability in the data has been added in the abstract to explain that scalability was not demonstrated for the scales: “Owing to low score variability between patients, the predefined threshold for scalability (≥0.40) was not reached for both scales.”.

3. Methods: A Mokken analysis of the two PDAS subscales (HAM-D6 and BPRS5) would be of interest given the results of the primary analysis (rather low coefficient of homogeneity).

The Loevinger coefficients of homogeneity derived from the Mokken analyses were 0.39 for HAM-D6 and 0.33 for BPRS5. They were lower than those observed in the original PDAS validation study (0.51 and 0.40 respectively) .

4. Methods: How were PDAS ratings "formatted" for the Mokken analysis (rescaled or)?

The instructions given in the original article were followed: “The BPRS item scores (from 1 to 7) were converted to a score between 0 and 4 according to this formula: (BPRS score – 1) x 2/3. For the Mokken analysis, which requires data on a numeric scale, the BPRS scores were converted as follows: 1 = 0, 2–3 = 1, 4-5 = 2, 6 = 3, and 7 = 4. In all analyses of the seven composite scales, the rating on Hamilton item 13 (general somatic) was multiplied by 2”.

We agree that the method section of our article was not clear enough and we therefore differentiated the method of calculating the total score in the presentation of the PDAS section and that used for the Mokken analysis in the statistical analyses section:

The PDAS instrument

“The total PDAS score is obtained by adding up the sum of the individual item scores after converting the BPRS item scores to a score between 0 and 4 with the formula (BPRS score – 1) x 2/3, and multiplying the score for item 13 on the Hamilton scale (general somatic) by 2.”

Statistical analyses

“(i) Scalability was assessed by Mokken analysis, generating a Loevinger coefficient of homogeneity for the PDAS total score. In the Mokken analysis, a Loevinger coefficient of homogeneity ≥0.40 is indicative of scalability [32]. For the present analysis the BPRS scores were converted as follows: 1 = 0, 2–3 = 1, 4-5 = 2, 6 = 3, and 7 = 4.”

5. Discussion: "Surprisingly, despite a highly significant result, the Mokken analysis did not confirm the scalability of the PDAS (Loevinger coefficient < 0.40). During the validation of the semi-structured PDAS interview, Köse Çinar and Østergaard [42] also failed to demonstrate the scalability of the PDAS (Loevinger coefficient 0.17) at baseline. They mentioned that it may be due to low variability in PDAS scores (22.8 ± 5.7)."

Yes - but at endpoint in the Köse Çinar and Østergaard study - the Loevinger coefficient was 0.45. This needs to be underlined (fits the explanation). I also think that it deserves mentioning that all four studies having previously assessed the scalability of the PDAS, have confirmed the scalability of the instrument:

https://pubmed.ncbi.nlm.nih.gov/23799875/

https://pubmed.ncbi.nlm.nih.gov/29501075/

https://pubmed.ncbi.nlm.nih.gov/25462426/

https://pubmed.ncbi.nlm.nih.gov/28535843/

This leads this reviewer to think, that the present sample of patients is "qualitatively different" from prior studies (see also comment 6B below).

This section has been completed as requested: “In general, studies assessing the scalability of the PDAS demonstrate its scalability [16,42,47] and even Köse Çinar and Østergaard found a Loevinger coefficient > 0.40 at endpoint of their study [41].”

6. Discussion: "Our data showed similar score with low variability (21.8 ± 4.9), compared to the original PDAS validation which showed higher score and variability (29.5 ± 11.3) [48]. These differences in PDAS score and variability may be due to the population. Despite the same inclusion criteria than the article of Østergaard [17], our population had lower HDRS score, more hallucinations, more suicide attempts and are all inpatients." Voir texte modifié

A: What is meant by "similar score" here?

The sentence has been changed to be clearer: “Our data showed a score with low variability (21.8 ± 4.9) close to that in their study [41] whereas the original PDAS validation study showed a higher score and variability (29.5 ± 11.3) [47].”

B: "These differences in PDAS score and variability may be due to the population. Despite the same inclusion criteria than the article of Østergaard [17], our population had lower HDRS score, more hallucinations, more suicide attempts"

This description leads this reviewer to think that there may be a fraction of the participants suffering from either a primary psychotic disorder (ICD-10: F20) or PTSD (as also suggested by one of the other reviewers).

Psychotic features in depressive disorder should be distinguished from delusional ideas and hallucinations that occur in schizophrenia, and flashbacks in PTSD should be distinguished from hallucinations that can occur in depressive disorder with psychotic features. Several other symptoms support the differential diagnosis between depressive disorder with psychotic features and schizophrenia or PTSD. As our patients were hospitalized in psychiatric units, they were assessed by hospital psychiatrists independent of the study who made a diagnosis during care. For recruitment, we specified that the patients who could be invited to participate in the study were those with a diagnosis of major depressive disorder with psychotic features. In addition, the study psychiatrist systematically checked the inclusion and exclusion criteria.

In our study, the hallucinations and the lower educational level of patients are correlated (please see attached table). This is consistent with the literature (Jääskeläinen E et al, 2018 ). In our study, 70 of the 94 patients were recruited from the psychiatric hospital centres of Clermont-Ferrand and Le Puy en Velay. These hospital centres treat patients with a lower educational level (only 17% of these patients spent more than 12 years in school) who often live in rural communities. The educational level of our patients was lower than the average educational level of patients in the studies of Meyers et al (2006) and SC Park et al (2014). This point has been emphasized in the conclusion: “However, other studies are needed to confirm the scalability of the PDAS and the generalisability of the results to a larger population.” 

Table: SAPS hallucination score according to educational level

Educational level SAPS hallucination score (mean ± SD)

< 12 years 2.3 ± 1.8

12 years 1.7 ± 1.9

> 12 years 1.1 ± 1.8

---

## [Decision Letter · Decision Letter 2]

8 Apr 2021

Cross-cultural evaluation of the French version of the Delusion Assessment Scale (DAS) and Psychotic Depression Assessment Scale (PDAS)

PONE-D-20-28115R2

Dear Dr. Jalenques,

We’re pleased to inform you that your manuscript has been judged scientifically suitable for publication and will be formally accepted for publication once it meets all outstanding technical requirements.

Kind regards,

Paolo Roma

Academic Editor

PLOS ONE

Reviewers' comments:

Reviewer's Responses to Questions

**Comments to the Author**

1. If the authors have adequately addressed your comments raised in a previous round of review and you feel that this manuscript is now acceptable for publication, you may indicate that here to bypass the “Comments to the Author” section, enter your conflict of interest statement in the “Confidential to Editor” section, and submit your "Accept" recommendation.

Reviewer #1: All comments have been addressed

Reviewer #2: All comments have been addressed

2. Is the manuscript technically sound, and do the data support the conclusions?

Reviewer #1: (No Response)

Reviewer #2: Yes

3. Has the statistical analysis been performed appropriately and rigorously? 

Reviewer #1: (No Response)

Reviewer #2: Yes

4. Have the authors made all data underlying the findings in their manuscript fully available?

Reviewer #1: (No Response)

Reviewer #2: Yes

5. Is the manuscript presented in an intelligible fashion and written in standard English?

Reviewer #1: (No Response)

Reviewer #2: Yes

6. Review Comments to the Author

Reviewer #1: (No Response)

Reviewer #2: All comments have been addressed.

......................................................................................................................................................................

7. PLOS authors have the option to publish the peer review history of their article (what does this mean?). If published, this will include your full peer review and any attached files.

Reviewer #1: No

Reviewer #2: No

---

## [Editor Report · Acceptance letter]

15 Apr 2021

PONE-D-20-28115R2 

Cross-cultural evaluation of the French version of the Delusion Assessment Scale (DAS) and Psychotic Depression Assessment Scale (PDAS) 

Dear Dr. Jalenques:

I'm pleased to inform you that your manuscript has been deemed suitable for publication in PLOS ONE. Congratulations! Your manuscript is now with our production department. 

Kind regards, 

on behalf of

Prof. Paolo Roma 

Academic Editor

PLOS ONE